# Enhancement and Optimization of Underwater Images and Videos Mapping

**DOI:** 10.3390/s23125708

**Published:** 2023-06-19

**Authors:** Chengda Li, Xiang Dong, Yu Wang, Shuo Wang

**Affiliations:** 1School of Electrical Engineering and Automation, Anhui University, Hefei 230601, China; z20201037@stu.ahu.edu.cn; 2State Key Laboratory of Management and Control for Complex Systems, Institute of Automation, Chinese Academy of Sciences, Beijing 100190, China; yu.wang@ia.ac.cn (Y.W.); shuo.wang@ia.ac.cn (S.W.)

**Keywords:** underwater image and video restoration, transmission map optimizer, saturation map, color correction

## Abstract

Underwater images tend to suffer from critical quality degradation, such as poor visibility, contrast reduction, and color deviation by virtue of the light absorption and scattering in water media. It is a challenging problem for these images to enhance visibility, improve contrast, and eliminate color cast. This paper proposes an effective and high-speed enhancement and restoration method based on the dark channel prior (DCP) for underwater images and video. Firstly, an improved background light (BL) estimation method is proposed to estimate BL accurately. Secondly, the R channel’s transmission map (TM) based on the DCP is estimated sketchily, and a TM optimizer integrating the scene depth map and the adaptive saturation map (ASM) is designed to refine the afore-mentioned coarse TM. Later, the TMs of G–B channels are computed by their ratio to the attenuation coefficient of the red channel. Finally, an improved color correction algorithm is adopted to improve visibility and brightness. Several typical image-quality assessment indexes are employed to testify that the proposed method can restore underwater low-quality images more effectively than other advanced methods. An underwater video real-time measurement is also conducted on the flipper-propelled underwater vehicle-manipulator system to verify the effectiveness of the proposed method in the real scene.

## 1. Introduction

The ocean is abundant in oil, combustible ice, fish stocks, and mineral resources, which has attracted the attention of more and more governments and researchers. Mainstream marine vehicles, such as underwater remotely operated vehicles (ROVs), autonomous underwater vehicles (AUVs), and commercial submarines, are equipped with deep-sea underwater imaging systems [1]. An underwater camera imaging system is a cost-effective resource that is used extensively because of its low cost, low technical requirements, easy transplantation, and convenient configuration compared with other techniques, such as multispectral stereoscopic imaging [2] and laser scanning imaging [3].

However, underwater images and video often display serious color cast, sharpness degradation, and contrast reduction affected by the complex underwater environment, absorption and scattering of light, suspended particles, and other factors [4]. Poor visibility brings inevitable problems in computer vision tasks, such as object recognition and object detection [5]. In a sense, how to restore underwater images and video is challenging and meaningful.

Existing underwater restoration methods are mostly based on the image formation model (IFM) [6,7,8], which describes the linear relationship among the direct component, the forward scattering component, and the backward scattering component. In this model, two crucial parameters, the background lights and the transmission map, were estimated to recover underwater haze-free images. He et al. [9] observed that in most local patches, the lowest pixel intensity of the R–G–B color channel based on land haze-free images is close to zero, and statistically summarized the dark channel prior (DCP), which was widely applied for underwater image restoration. Inspired and motivated by the imaging similarity between land-based images and underwater images, some typical image restoration methods based on the IFM and the DCP have been proposed. Chao et al. [10] directly applied the DCP to underwater images that can only handle the scattering problems effectively, but the problems of absorption were not solved. Considering the notable influence of red-light attenuation under most underwater circumstances on the dark channel, only the dark channel of the G–B color channels [11] was proposed to enhance underwater images, which improved the accuracy of the estimated transmission map. Galdran et al. [12] proposed a red channel compensation method as a variant of the DCP, which was described as the dark channel of the inverted R channel and G–B channel, and the pixel of the highest intensity in the R channel was used to estimate BL. Peng et al. [13] proposed a complex method based on image blurriness and light absorption (IBLA) to obtain more precise BL estimation and underwater scene-depth estimation. However, this method is quite time-consuming. Song et al. [14] proposed a new underwater dark channel prior (NUDCP), which included a BL estimation statistic model and the TM of the R–G–B channel estimation model combination of the compensation by the scene-depth map and the optimization by the saturation map to restore underwater images, but that parameter estimation was greatly affected by the environment. Liang et al. [15] proposed a backscattered light estimation that combined wavelength-dependent attenuation, image blurriness, and brightness priors based on a hierarchical searching technique.. Yu et al. [16] proposed a novel transmission map optimization model designed by Yin–Yang pair optimization to restore underwater images. This method can suppress background noise and enhance detailed texture features but simultaneously increases complexity and computing cost.

It is challenging to enhance and restore underwater images in distinct water types with different distortions. Available underwater image enhancement and restoration methods either cannot obtain the accurate BL and TM estimation or have low algorithm processing speed. To overcome these problems, we propose an effective and high-speed underwater image and video restoration method in this paper. The highlights of this paper are summarized as follows:An accurate and high-speed background light estimation method is proposed, which is not only suitable for distinct types of underwater images but also for low complexity. The proposed method is time-saving and adaptable for most underwater images.TM estimation with an improved optimizer is established. Integrating the compensation of the scene-depth map based on color attenuation prior (CAP) and the adaptive saturation map (ASM), an optimizer is designed to modify and refine the coarse TM. The proposed TM estimation method provides more accurate results and has lower complexity in different kinds of underwater images than other advanced models.An improved white balance (WB) algorithm is employed to improve the color cast and visibility for restored images. The gain factor is adaptively selected related to the restored underwater image intensity to avoid over- or under-correction.

The rest of this paper is organized as follows. The underwater image formation model and underwater image restoration based on the DCP are described in Section 2. In Section 3, the BL estimation and the design of the TM optimizer are depicted and explained in detail. The underwater image restoration method with color correction is presented. Section 4 reports the experimental results compared with other state-of-the-art methods. Section 5 discusses the results and conclusion.

## 2. Related Works

Underwater image restoration and degradation are mutually inverse operations when the light propagates in the water medium. Underwater image restoration based on the image formation model (IFM) is introduced briefly. The parameter estimations of global background lights and a transmission map are two vital parameters to restore underwater images based on the DCP, which can affect the results directly.

### 2.1. Underwater Image Formation Model

A simplified underwater image formation model is used to describe the haze image [17]:(1)$$I^{C}(x,y)=J^{C}(x,y)\;t^{C}(x,y)+B^{C}(1-t^{C}(x,y))$$
where *I^C^*(*x*,*y*) represents the image intensity captured by the camera from the scene at the position (*x*,*y*); *J^C^* refers to the corresponding haze-free image; *B^C^* denotes the background lights, which is a three-dimensional vector; *C* ϵ {*R*, *G*, *B*} denotes one of the R–G–B color channels; and *t^C^*(*x*,*y*) denotes the transmission map, and it is an exponential attenuation function related to the spectral volume attenuation coefficient *β*(*x*,*y*) and the scene depth *d*(*x*,*y*), which can be expressed as follows:(2)$$t^{C}(x,y)=e^{-\beta (x,y)\;d(x,y)}$$

### 2.2. Underwater Image Restoration Based on the DCP

Owing to the optical imaging similarities between land-based images and underwater images, the coarse transmission map (TM) is estimated based on the DCP under the statistical assumption *J*_dark_(*x*,*y*) = 0.

First, we apply the minimum value operator on a local patch Ω(*x*,*y*) on both sides in Equation (1):(3)$$\underset{z\in \Omega (x,y)}{\min}I^{C}(z)=\underset{z\in \Omega (x,y)}{\min}J^{C}(z)t^{C}(z)+\underset{z\in \Omega (x,y)}{\min}B^{C}(1-t^{C}(z))$$

Generally, the background light *B^C^* is assumed to be constant and positive, and both sides of Equation (3) are divided by *B^C^*. Then, *t^C^*(*z*) is constant and continuous on a small patch, so we have
(4)$$\frac{\underset{z\in \Omega (x,y)}{\min}I^{C}(z)}{B^{C}}=\frac{\underset{z\in \Omega (x,y)}{\min}J^{C}(z)}{B^{C}}t^{C}(z)+1-t^{C}(z)$$

Then, the minimum filter operator is applied among R–G–B color channels as follows:(5)$$\frac{\underset{C}{\min}\underset{z\in \Omega (x,y)}{\min}I^{C}(z)}{B^{C}}=\frac{\underset{C}{\min}\underset{z\in \Omega (x,y)}{\min}J^{C}(z)}{B^{C}}t^{C}(z)+1-\underset{C}{\min}t^{C}(z)$$

According to the dark channel prior, we get
(6)$$\frac{\underset{C}{\min}\underset{z\in \Omega (x,y)}{\min}J^{C}(z)}{B^{C}}=0$$

Combining Equation (5) with Equation (6), the multiplicative term is deleted and the coarse transmission map of the R color channel *t^R^*(*z*) is expressed as follows:(7)$$t^{R}(z)=1-\frac{\underset{C}{\min}\underset{z\in \Omega (x,y)}{\min}I^{C}(z)}{B^{C}}$$

Finally, the restored underwater haze-free image *J^C^*(*x*,*y*) is calculated from
(8)$$J^{C}(x,y)=\frac{I^{C}(x,y)-B^{C}}{\min(\max(t^{C}(x,y),0.1),0.9)}+B^{C}$$

The constants 0.1 and 0.9 are the lower and upper limits of the transmission map, respectively.

The coarse transmission maps of the R channel for typical underwater images are shown in Figure 1. The accurate R channel TMs based on the DCP are estimated for underwater images in Figure 1a–c, with uniform illumination and a gloomy foreground scene. On the contrary, in Figure 1d, the high-brightness target or white object near the camera is mistaken for the far scene, which causes the estimated transmission map of the region to be underestimated. In Figure 1e, because of some regions in which the red channel intensity is lower than the estimated background lights, such as the area behind the statue, the local TM is overestimated. Figure 1f shows the most inhomogeneous illumination scene in the shark back region and the poor TM estimation.

On the one hand, the TM based on the DCP can be estimated precisely when the illumination of the scene from the light source is from far away, which is regarded as uniform illumination, just contrary to the artificial light. On the other hand, the estimated BL is also inseparable from the accurate TM estimation. However, it is extremely difficult to estimate the BL and the TM accurately in most real underwater images. Like the last three salient cases in Figure 1, the transmission map estimation is inaccurate, which affects the subsequent image restoration.

## 3. Problem Formulation

To solve the above problems, an improved background light estimation and transmission map optimizer are proposed to restore underwater images from the aspects of visibility, contrast, and color. The flow-process diagram of the whole method is shown in Figure 2. First, the underwater image color channels are judged whether to be seriously attenuated, and an appropriate method is chosen to estimate the BL. Secondly, a TM optimizer of the R color channel is designed to modify and refine the coarse TM based on the DCP. Then, the TMs of the remaining G and B channels can be calculated by the attenuation coefficient ratios of their channels to the red channel relatively. Using the estimated BL and TM, we can obtain a haze-free underwater image. Finally, an adaptive color correction method is applied to adjust and correct the brightness and color of the restored image.

### 3.1. Background Light Estimation

Recently, various methods for estimating background light have been developed, such as DCP_RGB_ or its variants [11], the quad-tree decomposition searching method [5,18], the BL candidate selection region [13], and BL estimation based on the statistic model [14], which worked well only for given underwater images, as it mostly had high computational complexity and was time consuming. Affected by the type of water, the degree of different color channel attenuation is different. The red channel is attenuated more severely than the G–B channel in these images captured in the open ocean. In the coastal oceans, the blue channel is more attenuated than the others.

Inspired by the paper in [15], we used Formula (9) to determine whether some color channel is a severe attenuation channel by calculating the ratios between the maximum channel pixel average and the channel pixel average of the input image and chose the relative method to compute BL. The BL estimation based on the statistical model can obtain an accurate BL of the raw image with no or little red component rapidly, which is not suitable for other channels with serious attenuation. Therefore, we extended it to other attenuation color channels, as shown in Equation (10). For underwater images with severe attenuation or darker intensity, it is more accurate to utilize an improved BL model based on statistic model to estimate the BL. For other situations, the DCP_RGB_ method is utilized. Thus, we took full advantage of the advantages of these two methods to obtain
(9)$$BL=\left\{\begin{array}{l} BL1\begin{array}{ll} , & \max(Avg(I^{C}))/\min(Avg(I^{C}))\geq 2 \end{array}\\ BL2\begin{array}{rr} , & \end{array}otherwise \end{array}\right.$$

Here,
(10)$$BL1=\left\{\begin{array}{l} 1.13\times Avg^{m}+1.11\times Std^{m}-25.6,\begin{array}{c} Avg(I^{C})>\delta \end{array}\\ \frac{140}{1+14.4\times exp(-0.034\times Med^{n})},otherwise \end{array}\right.$$
(11)$$BL2=I^{C}(\arg\max\{\underset{\Omega (x,y)}{\min}(\underset{C}{\min}(I^{C}(x,y)))\})$$
where *δ* is a threshold and set *δ* = 64, which was consulted in the literature [15]; *m* denotes the color channel, with an average intensity value greater than *δ*; and *n* denotes the opposite channel. *Avg*, *Std*, and *Med* denote the average value, the standard deviation, and the median value of the channel in the underwater image *I*, respectively. To avoid producing over- or under-estimation, the BL value was limited to between 5 and 250. Therefore, the ultimate estimated BL is:(12)$$B^{C}=\min(\max(B^{m,n},5),250),C\in \{R,G,B\}$$

### 3.2. TM Optimizer Design

The design steps of the TM optimizer are depicted minutely in this part, which consists of two components: the correction of the scene-depth map and the optimization of the adaptive saturation map (ASM). For some special cases, our optimizer can eliminate and refine the unreasonable and error areas in the transmission map based on the DCP to obtain the accurate TM.

Firstly, the scene depth is estimated based on the color attenuation prior (CAP). Through the experiments on extensive foggy images, the scene depth can be described as a positively correlated function with the fog concentration and the difference value between image brightness and saturation [19]. The scene-depth estimation model is:(13)$$d(x,y)=\theta_{0}+\theta_{1}\;v(x,y)+\theta_{2}\;s(x,y)+\tau (x,y)$$
where *θ*_0_, *θ*_1_, and *θ*_2_ are coefficients; *τ* is a random parameter; *v*(*x*,*y*) is the image brightness; and *s*(*x*,*y*) is the image saturation. Those parameters can be calculated by supervised learning, and we directly followed the optimal solution in this paper [19]: *θ*_0_ = 0.121779, *θ*_1_ = 0.959710, *θ*_2_ = −0.780245, *τ* = 0.041337.

The scene-depth maps of the underwater images might not be consistent with the reality in the corresponding region with the objects or targets with high brightness. Thus, the minimum filter is applied to settle those questions as follows:(14)$$d_{f}(x,y)=\underset{\Omega (x,y)}{\min}d(x,y)$$

The transmission map (TM) of R–G–B color channels is easily obtained as follows:(15)$$t^{C}(x,y)=e^{\left(-\beta^{C}d_{f}(x,y)\right)},\;C\in \{R,G,B\}$$

Just the R channel is considered, and we set *β^R^* = 1 and obtain:(16)$$t_{d}^{R}(x,y)=e^{-d_{f}(x,y)}$$
whereas the estimated TM in Equation (7) is related to the scene, intensity, and detail information of the raw image, which produces inaccurate estimation in specific cases easily. For example, some close background areas with a small intensity value of the R channel overestimate the TM value and are considered foreground [14]. The color attenuation prior (CAP) supposes that the near background area with a small intensity value has low brightness and saturation, and the TM value is correspondingly small, which is used to compensate for and refine the TM. The modified TM is expressed as:(17)$$t_{{}^{m}}^{R}(x,y)=\min\{t^{R}(x,y),t_{d}^{R}(x,y)\}$$

Secondly, when the underwater environment lacks light, we usually use artificial light (AL) to increase the view field of the external scene. Therefore, we ought to eliminate the effect of AL. However, a region with high brightness cannot indicate the emergence of artificial light. We found that the saturation value of a region illuminated by AL is low in HSV color space [12,14]. The image saturation map is expressed as follows:(18)$$S(x,y)=\left\{\begin{array}{l} 1,\max I^{C}(x,y)=0\\ \frac{\max I^{C}(x,y)-\min I^{C}(x,y)}{\max I^{C}(x,y)},otherwise \end{array}\right.$$

Image saturation characterizes the vividness of color in an image, which can be used to reflect the effects of artificial lights. With an increase in white light, the channels lose saturation gradually, and the regions illuminated by artificial light force the brightness of pixels in the three channels to be closed, which results in the saturation value being very low. To some extent, the areas with low saturation in the image can be considered the areas illuminated by AL. Thus, we defined an adaptive saturation map (ASM) to describe this phenomenon as follows:(19)$$S_{p}(x,y)=1-\alpha \;S(x,y)$$
where *α* is the coefficient related to the mean value of the saturation map. We applied ASP to modify and optimize the TM to abate the intensity of the inhomogeneous illumination or artificial light region. Relevantly, the intensity in other areas has little impact.
(20)$$t_{f}^{R}(x,y)=\max\{t_{m}^{R}(x,y),S_{p}(x,y)\}$$

Applying our optimizer to modify and refine TM, the inaccurate TMs of Figure 1d–f can be corrected and refined. Then, guided filtering [20] is applied to preserve the edges. The above inaccurate TMs are refined by our TM optimizer, as shown in Figure 3.

In Equation (2), the transmission map needs to be calculated on each color channel, and three channel transmission maps are not disrelated. Only one metric and two scale coefficients [21] need to be computed as follows:(21)$$\left\{\begin{array}{l} t_{f}^{R}(x,y)=e^{-\beta^{R}d(x,y)}\\ t_{f}^{G}(x,y)=e^{-\beta^{G}d(x,y)}={\left(e^{-\beta^{R}d(x,y)}\right)}^{\frac{\beta^{G}}{\beta^{R}}}={\left(t_{f}^{R}(x,y)\right)}^{{}^{\frac{\beta^{G}}{\beta^{R}}}}\\ t_{f}^{B}(x,y)=e^{-\beta^{B}d(x,y)}={\left(e^{-\beta^{R}d(x,y)}\right)}^{{}^{\frac{\beta^{B}}{\beta^{R}}}}={\left(t_{f}^{R}(x,y)\right)}^{{}^{\frac{\beta^{B}}{\beta^{R}}}} \end{array}\right.$$

Since the TM is estimated on a small pitch Ω(*x*,*y*), the transmission map *t^C^*(*x*,*y*) easily demonstrates some color halos and block artifacts. Finally, the guided filter [20] is applied to solve these problems.

### 3.3. Color Correction

Although our method can effectively remove the fog in underwater images, some new problems also appear, such as the restored images having poor contrast, uneven color, and low brightness, which makes it hard to obtain valuable information. Inspired by the white balance algorithm [21], an improved color correction is proposed. Compared to the original method, its gain factor is selected automatically based on the brightness of the input image and can be expressed as:(22)$$\left\{\begin{array}{l} I_{out}=\frac{I_{in}}{V_{p}\times (m/m_{ref})+\lambda_{as}}\\ m_{ref}=\sqrt{{\left(m_{R}\right)}^{2}+{\left(m_{G}\right)}^{2}+{\left(m_{B}\right)}^{2}} \end{array}\right.$$
where *I_in_* and *I_out_* denote the input restored image and the output corrected image; *m_R_*, *m_G_*, and *m_B_* denote the mean value of the image *I_in_* color channels; and *V_p_* is the maximum intensity value of the image *I_in_*. The parameter *λ_as_* is the gain factor to adjust the color of the input image, whose range is 0 to 0.5. According to the color correction effect of the *λ_as_* selection for the restored image, an auto-select gain factor mothed is proposed to obtain the desired *λ_as_*. Our principle is as follows: When the maximum value of *m_R_*, *m_G_*, and *m_B_* is greater than 0.45, the image *I_in_* is brighter and *λ_as_* is close to 0.5; in other words, if the maximum value is less than 0.45, the image is dark and *λ_as_* is closed to 0, which can be expressed as:(23)$$\lambda_{as}=\left\{\begin{array}{l} 0.5\times \tanh(m_{2}/m_{1})\begin{array}{rr} , & m_{2}>0.45 \end{array}\\ 0.5\times \tanh(m_{1}/m_{2})\begin{array}{rr} , & otherwise \end{array} \end{array}\right.$$
where *m*_2_ and *m*_1_ are the maximum value and the minimum value of *m_R_*, *m_G_*, and *m_B_*. The *tanh*(·) represents the hyperbolic tangent function.

Some typical underwater images restored by our method are shown in Figure 4 to explain the whole process of image restoration. Figure 4b,c represent the coarse TM based on DCP and the precise TM refined by the TM optimizer. Figure 4d shows the restored image without color correction. Figure 4e denotes the final enhanced image.

## 4. Results and Evaluation

In this section, we analyze and evaluate the proposed underwater image restoration and enhancement method qualitatively and quantitatively compared with other state-of-the-art methods. All tested underwater images were from the underwater benchmark dataset [22] and all codes were from open-source codes, which were run on a Windows 11 PC with AMD Ryzen 9 5900HX with Radeon Graphics@3.30 GHZ, 6.00 GB, running Python 3.6.0.

### 4.1. Evaluation of Objectives and Approaches

The experiments were designed to verify the effectiveness and performance of our method from the following three perspectives:(1)To prove the effectiveness of the TM optimizer;(2)To prove the comprehensive performance of the proposed method;(3)To test the real-time performance of underwater video.

In experiment (1), we implemented the evaluations by comparing them with other classical and representative underwater image restoration methods, which include the dark channel prior (DCP), the maximum intensity prior [23] (MIP), the underwater dark channel prior (UDCP), the image blurriness and light absorption (IBLA), the underwater light attenuation prior [24] (ULAP), and a new underwater dark channel prior (NUDCP).

To make the experimental results more convincing, four image-quality indexes were employed, which were as follows: two full-reference (FR) indexes (the peak signal-to-noise ratio (PSNR) and the structural similarity (SSIM)) and two non-reference (NR) indexes (the underwater image-quality measure [25] (UIQM) and the blind referenceless image spatial-quality evaluator [26] (BRISQUE)).

The PSRN index mainly measures the average ratio of signal maximum energy to signal noise energy and is widely used. A greater value indicates higher similarity. The SSIM index describes the structural similarity information between the enhanced image and the contrast reference image intuitively. A higher SSIM value represents better performance. Although the PSNR and SSIM indexes need contrast reference images, which are difficult to obtain, they can reflect the influence of the added noise of underwater restored images on the structure of the contrast reference image. The UIQM index takes colorfulness, sharpness, and contrast as the measurement components and combines the components linearly to assess image quality comprehensively. A higher value represents better visibility and quality. The BRISQUE index can measure the naturalness losses in underwater images. A lower value indicates a better result.

### 4.2. Performance of Transmission Map Optimizer for Single Underwater Image

First, the performance of our TM optimizer was verified with different kinds of underwater images, which included greenish and bluish underwater images. To express our performance objectively, we chose various kinds of restored underwater images compared with other existing methods shown in Figure 5 and Figure 6, and their quantitative image-quality assessment indexes are recorded in Table 1, Table 2, Table 3 and Table 4.

Figure 5 and Figure 6 show the restored results obtained by various transmission map estimation methods for two greenish underwater images and two bluish underwater images, respectively. Underwater images are commonly greenish and bluish owing to the light absorption and scattering, which lose valuable details and cause color deviation. For those underwater images, our restored approach not only met the requirement of improving contrast but also removed the color cast.

In Figure 5(1,2), the UDCP method caused more serious color distortion and a darker background than the raw image. The DCP and MIP methods almost had no effect on improving contrast and color deviation. We noticed that the restored underwater images in Figure 5(1,2) by the IBLA, ULAP, and NUDCP methods seemed to have similar results in either improving the contrast or eliminating the degraded color, and the restored images were still low contrast and greenish, which was confirmed by the quantitative PSNR, SSIM, UIQM, and BRISQUE indexes from Table 1 and Table 2. The contrast was improved evidently and the color distortion was removed in Figure 5(1f,2f), and the restored results were more realistic than others. Although the UIQM index wasn’t the best, as shown in Table 2, the other indexes were sufficient to prove the effectiveness of the proposed TM optimizer.

The image in Figure 6(3) is bluish and dark. From the restored results, we found that DCP, UDCP, MIP, and ULAP made the image less bright and a more serious bluish tone, and the SSIM index and UIQM index were low, which indicates that they produced little improvement. The IBLA and NUDCP methods improved brightness and contrast, but they were still insufficient compared with our method. The quantitative indicators of our method were still beaten slightly. For the slight bluish image in Figure 6(4), the DCP, IBLA, and ULAP methods still retained bluish illumination, which caused litter de-hazing and a restored effect. The MIP method not only did not improve the color distortion but also aggravated it. The quantitative evaluation shows that the proposed TM optimizer yielded the highest values of the PSNR, SSIM, UIQM, and BRISQUE indexes, which indicates that our method produced a more gratifying and effective performance than others.

### 4.3. Comprehensive Performance for Single Underwater Image

Our proposed algorithm aims to enhance the contrast, eliminate color deviation, and improve the visibility of the underwater image. Thus, the proposed color correction method was carried out on the restored images as post-processing to restore underwater images overall.

To evaluate the performance of the proposed method comprehensively, we compared with the DCP, UDCP, MIP, IBLA, ULAP, and NUDCP. These methods add histogram equalization (HE) to correct the color of restored and enhanced images. Since the NUDCP method designed its color correction algorithm, we continued to employ it and meanwhile named it NU_CC. For convenience, the proposed method without color correction was named O_WCC, and the overall restoration was called “ours”.

The quantitative evaluation and data indexes are recorded in Table 5, showing the average values of 100 raw images, which were picked randomly from the underwater benchmark datasets. Figure 7 and Figure 8 show two kinds of typical underwater image restoration results. From Table 5, we discovered that after they were handled by the HE algorithm, the DCP, UDCP, MIP, and IBLA methods produced distinct and valuable improvements in image quality; however, sometimes it was just the opposite.

The overall restoration results for the underwater images restored by different methods can be seen in Figure 7 and Figure 8. To increase the comparison, two opposite results were selected to illustrate the necessity of color correction. The DCP, UDCP, MIP, and IBLA methods often failed to eliminate the greenish and bluish color in Figure 7b–e and Figure 8b–e. Color correction can effectively improve brightness, remove the color cast, and enhance contrast. However, after being enhanced by HE, as shown in Figure 7h–l and Figure 8h–l, the restored underwater images demonstrated high brightness and their colors were overcorrected and oversaturated, which masked significant information in the images. Although the color correction method can improve the UIQM index value of the restored images, as shown in Table 5, the unnatural images did not satisfy our requirements. Our method could effectively overcome the image degradation to obtain the haze-free underwater images shown in Figure 7g and Figure 8g. The final image was enhanced and restored by our improved white balance method, shown in Figure 7n and Figure 8n, considering the uniqueness and correlation of each color channel and image brightness, and were neither oversaturated nor over-enhanced, which is very consistent with human vision.

Lastly, we selected some different underwater images from 60 challenging underwater images, including greenish underwater images, bluish underwater images, turbid underwater images, dark underwater images, and low-visibility underwater images. The ultimate results of restoration by different enhancement and restoration methods are shown in Figure 9 to show the overall performance of our proposed method.

### 4.4. Enhancement for Underwater Video

Our method is intended to carry out underwater or undersea image and video restoration on underwater mobile systems, such as AOVs and UAVs [27,28]. In general, the hardware resource and computational ability of those devices show pretty poor performance, so the component of real time is challenging and significant. We tested our proposed underwater image restoration method in real time on a flipper-propelled underwater vehicle-manipulator system [29], and the size of the underwater R–G–B image obtained by the system webcam was 312 pixels × 554 pixels. In the marine environment, a two-minute underwater video with 3000 frames was enhanced and restored by different methods, which included the DCP, UDCP, MIP, ULAB, NUDCP, and ours. The real-time measurement results are listed in Table 6.

The DCP, UDCP, MIP, and ULAP were only applied to enhance and restore a single image, and their mean processing speed for underwater video was much slower and could not be used for real-time enhancement and restoration. The NUDCP method could improve the speed of underwater video enhancement, but our processing time was 90 ms–100 ms per underwater image and the average restoration speed was more than 10 FPS, which is nearly four times as fast as the NUDCP because some functions were optimized and the parameters could be solved quickly. Our method can improve the processing speed of underwater video restoration greatly and can be applied to engineering tasks in real time.

## 5. Conclusions

This paper proposes an efficient underwater image and video restoration method that contains a fast and practical background light estimation method, an accurate transmission map estimation based on a TM optimizer, and an adaptive color correction algorithm. The designed method can be applied to a variety of environments and can meet real-time requirements. The quality of the restored underwater image and video depends on the accuracy of the estimated BL and the TM. Meanwhile, the BL and the TM are also closely connected, and the accuracy of the BL influences the TM estimation. Thus, an accurate BL estimation method is particularly significant. To verify the performance and the real time of our proposed method for underwater image and video both qualitatively and quantitatively, we designed several corresponding experiments to explain and demonstrate.

The proposed method performed better at enhancing underwater image and video. Because the background light cannot be estimated completely and accurately under special circumstances, such as a scene with generous artificial light and lots of sediment, our method may obtain the wrong transmission map, which makes the restored images have a color imbalance. For most attenuated underwater images, our method can obtain a better result. This will be instrumental in improving the visual perception ability of underwater vehicles.

We will continue to study how to accurately estimate the BL and optimize the TM for complex underwater environments in the future. Meanwhile, future research will pay attention to the real-time processing speed of underwater images and video to meet the physical demands of GPU programming to provide a visual basis for complex underwater operations.

## Figures and Tables

**Figure 1 sensors-23-05708-f001:**
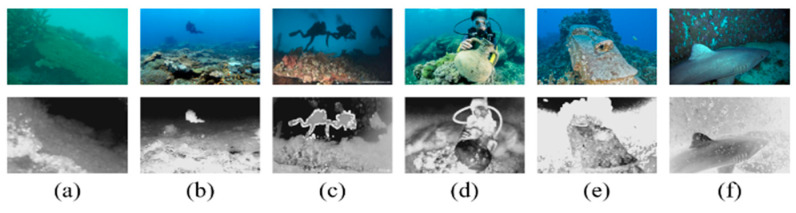
Several classical underwater images and their TM of the R color channel based on the DCP. The upper layer is the original image, and the lower layer is their corresponding transmission map. (**a**–**f**) represent different underwater images.

**Figure 2 sensors-23-05708-f002:**
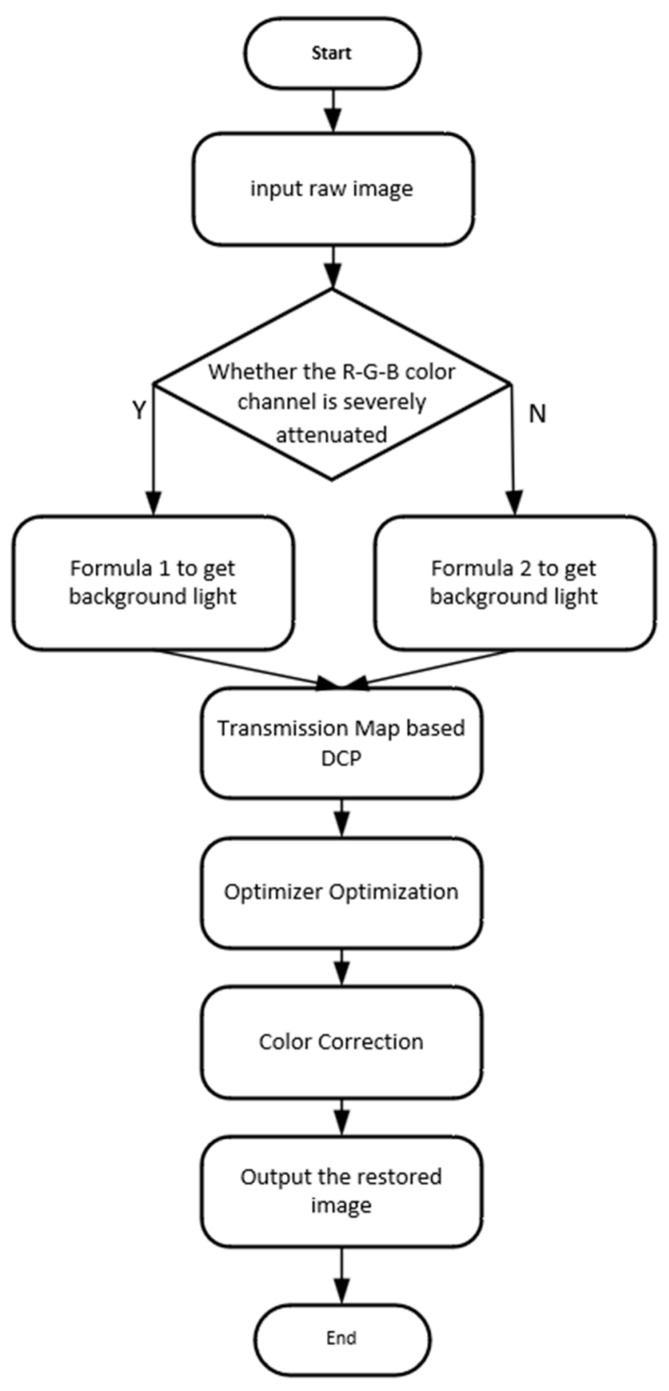
Flowchart of the proposed method.

**Figure 3 sensors-23-05708-f003:**
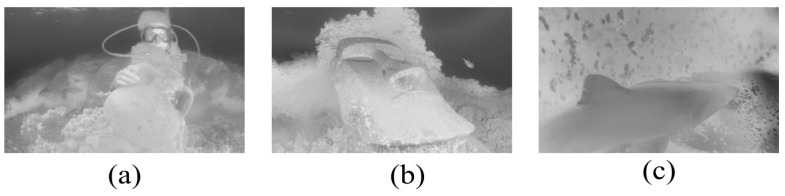
The above inaccurate TMs of the R channel are refined with our optimizer for Figure 1d–f. (**a**) The underestimated TM for the clay pot in the foreground region is accurately corrected; (**b**) the overestimated TM for the back of the statue in the background region is accurately corrected; (**c**) the TM in the shark back area with the non-uniform illumination is improved.

**Figure 4 sensors-23-05708-f004:**
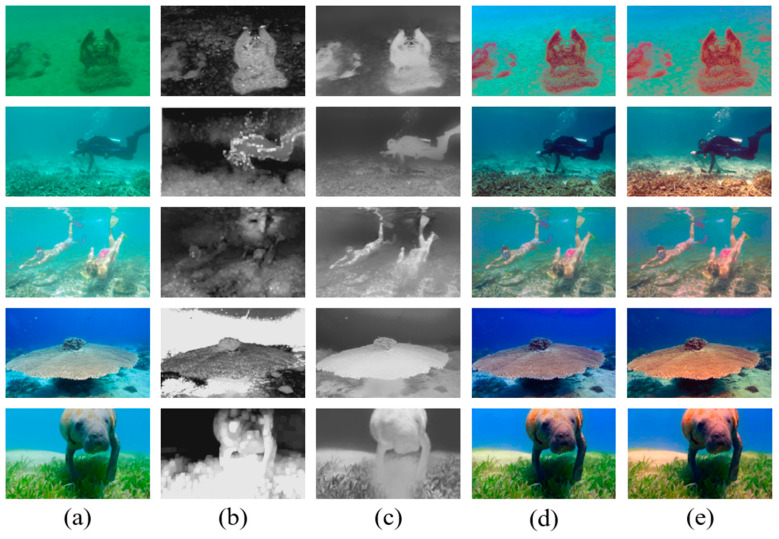
The whole processing of the proposed underwater image restoration method. (**a**) Raw image; (**b**) the TM of the R channel based on DCP; (**c**) refinement of the TM optimized by our TM optimizer; (**d**) the restored image; (**e**) the enhanced image with color correction.

**Figure 5 sensors-23-05708-f005:**
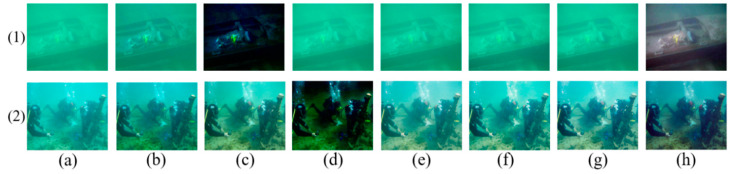
Image restoration results with different TM obtained for the greenish images. (**a**) Raw images; (**b**) DCP; (**c**) UDCP; (**d**) MIP; (**e**) ILBA; (**f**) ULAP; (**g**) NUDCP; (**h**) ours.

**Figure 6 sensors-23-05708-f006:**
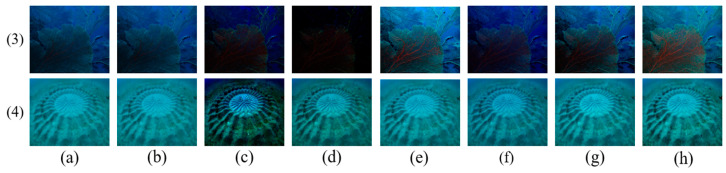
Image restoration results with different TM obtained for the bluish images. (**a**) Raw images; (**b**) DCP; (**c**) UDCP; (**d**) MIP; (**e**) ILBA; (**f**) ULAP; (**g**) NUDCP; (**h**) ours.

**Figure 7 sensors-23-05708-f007:**
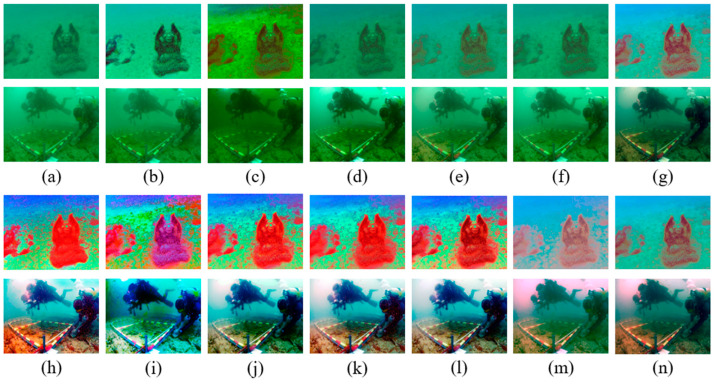
Comparative results for greenish images. (**a**) Raw images; (**b**) DCP; (**c**) UDCP; (**d**) MIP; (**e**) IBLA; (**f**) NUDCP; (**g**) O_WCC; (**h**) DCP + HE; (**i**) UDCP + HE; (**j**) MIP + HE; (**k**) IBLA + HE; (**l**) ULAP + HE; (**m**) NU_CC; (**n**) ours.

**Figure 8 sensors-23-05708-f008:**
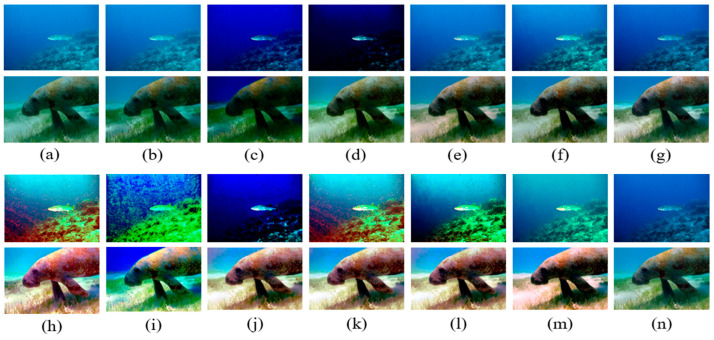
Comparative results for bluish images. (**a**) Raw images; (**b**) DCP; (**c**) UDCP; (**d**) MIP; (**e**) IBLA; (**f**) NUDCP; (**g**) O_WCC; (**h**) DCP + HE; (**i**) UDCP + HE; (**j**) MIP + HE; (**k**) IBLA + HE; (**l**) ULAP + HE; (**m**) NU_CC; (**n**) ours.

**Figure 9 sensors-23-05708-f009:**
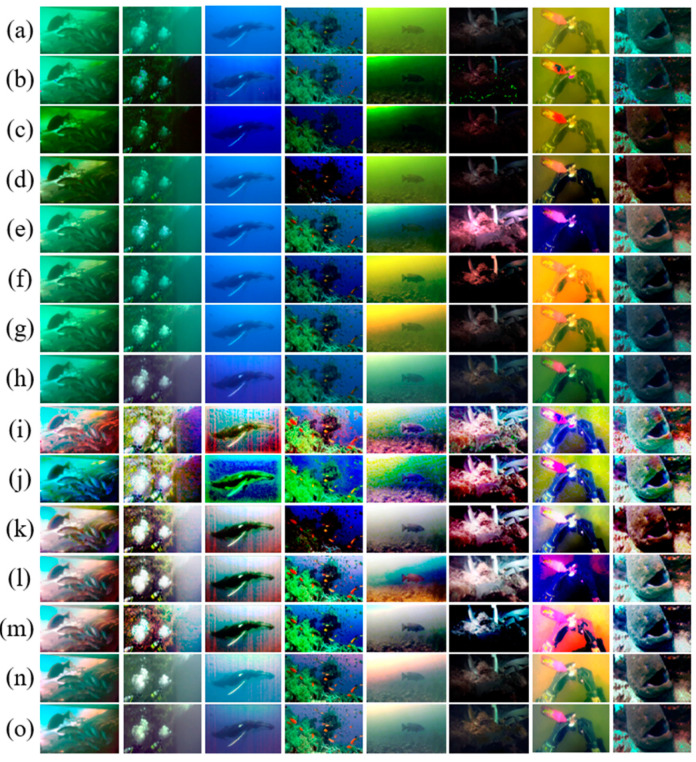
Comparative results for challenge underwater images. (**a**) Raw images; (**b**) DCP; (**c**) UDCP; (**d**) MIP; (**e**)IBLA; (**f**) ULAP; (**g**) NUDCP; (**h**) O_WCC; (**i**) DCP + HE; (**j**) UDCP + HE; (**k**) MIP + HE; (**l**) IBLA + HE; (**m**) ULAP + HE; (**n**) NU_CC; (**o**) ours.

**Table 1 sensors-23-05708-t001:** Quantitative analysis of Figure 5(1).

Methods	Indexes			
	PSNR	SSIM	UIQM	BRISQUE
DCP	15.65	0.652	0.78	49.56
UDCP	17.66	0.251	0.35	44.56
MIP	18.53	0.734	1.21	44.27
IBLA	22.59	0.762	1.17	42.88
ULAP	23.96	0.759	1.27	41.18
NUDCP	25.54	0.763	1.29	42.57
Ours	**28.63**	**0.896**	**1.72**	**39.32**

The bold is to highlight the best indicator results.

**Table 2 sensors-23-05708-t002:** Quantitative analysis of Figure 5(2).

Methods	Indexes			
	PSNR	SSIM	UIQM	BRISQUE
DCP	18.32	0.676	1.31	18.73
UDCP	16.81	0.314	0.77	19.08
MIP	15.45	0.815	**1.68**	25.45
IBLA	23.57	0.786	1.59	15.37
ULAP	25.05	0.763	1.49	15.06
NUDCP	25.19	0.825	1.61	16.34
Ours	**27.99**	**0.852**	1.46	**14.65**

The bold is to highlight the best indicator results.

**Table 3 sensors-23-05708-t003:** Quantitative analysis of Figure 6(3).

Methods	Indexes			
	PSNR	SSIM	UIQM	BRISQUE
DCP	16.15	0.354	0.73	61.81
UDCP	17.06	0.237	0.82	58.92
MIP	19.95	0.058	0.59	57.25
IBLA	23.09	0.501	1.19	53.35
ULAP	24.05	0.297	0.87	55.69
NUDCP	27.02	0.401	1.07	54.32
Ours	**28.24**	**0.651**	**1.31**	**52.81**

The bold is to highlight the best indicator results.

**Table 4 sensors-23-05708-t004:** Quantitative analysis of Figure 6(4).

Methods	Indexes			
	PSNR	SSIM	UIQM	BRISQUE
DCP	18.28	0.575	1.09	45.62
UDCP	16.71	0.548	1.09	42.38
MIP	20.01	0.609	1.14	42.83
IBLA	24.43	0.628	1.19	39.53
ULAP	26.24	0.611	1.14	39.58
NUDCP	28.58	0.625	1.22	39.95
Ours	**28.99**	**0.736**	**1.41**	**38.62**

The bold is to highlight the best indicator results.

**Table 5 sensors-23-05708-t005:** Quantitative analysis of underwater image enhancement and restoration based on different methods.

Methods	Indexes			
	PSNR	SSIM	UIQM	BRISQUE
DCP	18.86	0.37	0.99	45.33
UDCP	17.79	0.39	0.82	43.91
MIP	20.81	0.52	1.02	38.89
IBLA	22.59	0.65	1.38	34.77
NUDCP	25.39	0.71	1.46	29.14
O_WCC	28.54	0.77	1.61	36.67
DCP + HE	19.08	0.41	1.58	38.96
UDCP + HE	19.43	0.49	1.63	38.91
MIP + HE	21.98	0.62	1.74	35.05
IBLA + HE	22.66	0.71	1.79	32.11
ULAP + HE	24.66	0.68	**1.81**	30.69
NU_CC	26.04	0.75	1.68	28.63
Ours	**28.75**	**0.85**	1.75	**27.67**

The bold is to highlight the best indicator results.

**Table 6 sensors-23-05708-t006:** The processing speed of enhancement and restoration for underwater videos.

	Test Time (s)	TPF (ms)	FPS
DCP	975	982	1.0
UDCP	2169	723	1.4
MIP	4758	1586	0.6
ULAP	2016	672	1.5
NUDCP	1155	385	2.6
Ours	**288**	**96**	**10.4**

*TPF*: time per frame, *FPS*: frame per second. The bold is to highlight the best indicator results.

## Data Availability

Not applicable.

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
