# Peer review of "Enhancement and Optimization of Underwater Images and Videos Mapping"

_sensors, 2023, doi:10.3390/s23125708_

Round 1
Reviewer 1 Report
An effective and fast-speed enhancement and restoration method based on the dark channel prior (DCP) for underwater images and video is presented in this manuscript. Several typical image quality assessment indexes are employed to testify that the proposed method can restore underwater low-quality images more effectively than other advanced methods.
After a careful review of the paper, I am suggesting the following comments to the authors to revise the manuscript.
1. The flowchart shown in Fig. 2 is in no-standard form. it is should be redrawn according to the research process.
2. In line 185, “δ = 64”, the basis for the selection should be stated.
Author Response
We have answered all of your questions

Reviewer 2 Report
The paper presents an effective method for enhancing and restoring underwater images and videos based on the dark channel prior (DCP). While the proposed method shows promise, there are some areas that need improvement before acceptance.
- The paper is well-organized but would benefit from clearer descriptions of the proposed method and technical details. Providing a flowchart or diagram of the overall process would enhance clarity.
- Improved Background Lights Estimation: More information is needed on the specific improvements made and how they overcome the limitations of previous methods.
- Provide a more thorough explanation of this algorithm, including techniques or equations used. Visual examples or comparisons would be valuable.
- Conclusion: Expand the conclusion section to summarize contributions, advantages, limitations, and future research directions.
- Use consistent terminology throughout the paper.
- Proofread for grammatical errors and improve clarity.
- Consider citing recent relevant works in the introduction, for example
i. https://doi.org/10.1016/j.apm.2023.02.004
ii. doi: 10.3934/era.2023137
iii. https://doi.org/10.3390/jmse10030360
Minor Revision
Author Response
We have answered your questions

Reviewer 3 Report
Dear Authors, you did interesting work and reported elaborately about it. Your line of thought in the paper is well written but misses details. By comparison of your work with other methods, you report excellent results. Hence, your method is advantageous over other methods and you pinpoint strengths and weaknesses.
You do not explain sufficiently the origin of your formalisms. The literature is referred to as text, but you do not refer to papers for the mathematical/programming side of your work. An exception seems to be the references to [19, 20] between Lines 240 – 265. Because of this lack, it seems to me ungrounded that your results are so good, as reported in various Tables.
I will plea to the editor for publication of your work if you improve on your text.
Below I give details about typos and a few critiques on your text.
1. The title misses a determiner in “… Optimizer of the Transmission Map”. If you do not prefer this, in plural you can omit determiners, then please make it plural: “… Optimizer of Transmission Maps”.
2. Your chosen title of the paper is long. Long titles do not get as many references as short titles (as reported by the publisher Elsevier). Is it an option for you to shorten the title? For instance in 9 words:”. Enhancement and Optimization of Underwater Images and Videos Mapping”.
3. In formulas (1) and (3) a small blank should be inserted before the first occurrence of the variable t. Currently the closing bracket clobbers with the t as follows: )t .
4. In formula (2) a small blank should be inserted before the first occurrence of the variable d.
5. Lines 115, ‘background light BC is homogeneous’ is this correctly formulated? Should it be ‘background light BC is assumed constant’?
6. Line 116-117, Formula (4) is redundant, so please delete and replace the words “as follows:” at Line 116 by ‘then’, to continues the sentence with line 118.
7. Line 126, what happened to the minimum operator of t at the left hand side of the equation? In (6) it is still present.
8. Line 126, you inserted extra parentheses around I(z). This does not correspond to formula (6). Please correct.
9. Line 127, says that you calculate I. But this is not true, the formula 128 calculates J given I. Please correct.
10. Line 129, You do not explain the constants in (9). Where do they come from, are they gauged in the literature?
11. Line 173, you say here ‘formula (10)’. You better say here ‘decision rule’.
12. Line 178, you say ‘Eq. (11)’. This again is a decision rule.
13. Line 201, has a wrong variable v. You should not use the Greek symbol here, because in Line 202 you seem to use the Arabic symbol v.
14. Line 201, has the X symbol for multiplication, while there is no vector here. Why not discard the multiplication sign here, as you did in formulas (10) and (2)?
15. Line 235, also has a redundant X symbol. This symbol is only needed if you have vector multiplication or decimal number representation such as in formula (11).
16. The formulas (9) and (11) hold empirical constants. To me, it seems that you do not fully explain the sources of the constants. It seems that this is missing. Which references?
17. Line 251, ‘matric’ is wrong. Do you mean ‘metric’ here?
18. Line 428, here you use RGB, stead of R-G-B elsewhere.
19. Line 458-459, this is wrong: ‘accurately under special environments …’ it should read ‘‘accurately in special environments …’, or you might prefer ‘accurately under special circumstances …’.
20. Line 460, ‘makes the restored images color imbalance.’ should have a past tense: ‘makes the restored images color imbalanced.’
21. Line 486, abbreviations used in this manuscript are not fully listed here: Lines 298 – 307 suddenly have new bracketed abbreviations. Why? MIP, NU_CC and others are missing. Isn’t better to put the abbreviations list earlier in the paper? For instance, at the end of the Introduction?
22. Is the literature complete? Do I miss correctly reference to recent papers on the matter, such as:
https://www.mdpi.com/2227-7390/11/6/1382
https://www.mdpi.com/2077-1312/11/5/949
https://www.mdpi.com/2077-1312/10/10/1513
https://www.mdpi.com/2072-4292/14/17/4297
https://www.frontiersin.org/articles/10.3389/fmars.2022.1024339/full
See remarks above.
Author Response
We have answered all of your questions
